# Long-term benefit of vasodilating beta-blockers in acute myocardial infarction patients with mildly reduced left ventricular ejection fraction

Ki Yung Boo[1,2☯], Miyeon Kim[1,2☯], Jae-Geun Lee [1,2*], Geum Ko[1], Joon Hyouk Choi[1,2],
Song-Yi Kim [1,2], Seung-Jae Joo [1,2], Jin-Yong Hwang [3], Seung-Ho Hur[4],
Kwang Soo Cha[5], Myung Ho Jeong [6] on behalf of the KAMIR-NIH registry investigators[¶]

1 Department of Internal Medicine, Jeju National University Hospital, Jeju, Republic of Korea,
2 Department of Internal Medicine, Jeju National University College of Medicine, Jeju, Republic of Korea,
3 Department of Internal Medicine, Gyeonsang National University School of Medicine, Gyeongsang National University Hospital, Jinju, Republic of Korea, 4 Keimyung University Dongsan Medical Center, Cardiovascular Medicine, Daegu, Republic of Korea, 5 Pusan National University Hospital, Busan, Republic of Korea, 6 Department of Internal Medicine, Chonnam National University Hospital, Gwangju, Republic of Korea

☯ These authors contributed equally to this work.
¶ A list of KAMIR-NIH registry investigators is provided in the Acknowledgment.
* tedljg@naver.com

## Abstract

Beta-blockers have been considered the cornerstone of treatment for patients with acute myocardial infarction (AMI). However, long-term benefits of vasodilating beta-blockers remain uncertain. This study aimed to investigate the long-term clinical benefits of vasodilating beta-blockers compared to conventional beta-blockers in AMI patients with mildly reduced ejection fraction (mrEF). Among 13,624 patients who enrolled in the nationwide AMI database of South Korea, the KAMIR-NIH Registry, 2,662 AMI patients with mrEF, who were prescribed beta-blockers at discharge were selected for this study. The primary outcome was a composite of cardiac death, recurrent MI, or hospitalization for heart failure (HF) during 3-year follow up period. In the entire cohort, the use of vasodilating beta-blockers at discharge was associated with lower incidence of primary outcome at 3-year (hazard ratio [HR] 0.80; 95% confidence interval [CI], 0.62–0.98; P = 0.039) compared to the use of conventional beta-blockers at discharge. In the propensity score–matched (PSM) cohort, the use of vasodilating beta-blockers at discharge was also associated with a significantly lower incidence of primary outcome (HR, 0.66; 95% CI, 0.50–0.88; P = 0.004) compared to the use of conventional beta-blockers at discharge. Furthermore, in the PSM cohort, the use of vasodilating beta-blockers was associated with lower incidences of the cardiac death (HR, 0.60; 95% CI, 0.39–0.92; P = 0.020), hospitalization for HF (HR, 0.72; 95% CI, 0.46–0.98; P = 0.042), and all-cause death (HR, 0.67; 95% CI, 0.48–0.93; P = 0.017) compared to the use of conventional beta-blockers. However, no significant differences were observed between the groups in the incidences of

**Data availability statement:** All relevant data are within the paper and its Supporting information files.

**Funding:** This work was supported by a research grant from Jeju National University Hospital development fund in 2022. The funders had no role in study design, data collection and analysis, decision to publish, or preparation of the manuscript.

**Competing interests:** The authors have declared that no competing interests exist.

recurrent MI (HR, 0.62; 95% CI, 0.34–1.14; $P = 0.122$), any revascularization (HR, 1.04; 95% CI, 0.76–1.42; $P = 0.821$), stroke (HR, 0.84; 95% CI, 0.44–1.60; $P = 0.589$), stent thrombosis (HR, 1.12; 95% CI, 0.40–3.11; $P = 0.833$). In AMI patients with mrEF, the use of vasodilating beta-blockers at discharge was associated with better long-term clinical outcomes compared to the use of conventional beta-blockers.

## Introduction

The beta-blockers have traditionally served as a cornerstone in treatment for patients experiencing acute myocardial infarction (AMI). Following AMI, beta-blockers demonstrate significant benefits, including reductions in blood pressure, myocardial oxygen demand, thrombosis, and potentially life-threatening arrhythmias, thereby solidifying their role as an essential therapeutic strategy in these patients.[1–5] The enduring reputation of beta-blockers as a fundamental treatment for AMI patients stems from these beneficial attributes.[6–9] However, it is important to note that a substantial portion of these recommendations originated from studies conducted in the pre-reperfusion era through randomized trials.

In the reperfusion era, the clinical efficacy of beta-blockers in improving clinical outcomes for AMI patients has not been consistently validated through prospective randomized investigations. The beta-blockers have shown substantial clinical benefits in reducing mortality and recurrent MI in patients with AMI and left ventricular (LV) systolic dysfunction, particularly those with LV ejection fraction (LVEF) of less than 40%.[5] However, the efficacy of beta-blockers for AMI patients with LVEF greater than 40%, particularly those who underwent successful primary percutaneous coronary intervention (PCI) following AMI, remains unclear. This ambiguity is further exacerbated in patients with an LVEF range of 41–49%, who may be classified as having either diminished or preserved LV function. Recently, the term "mildly reduced EF (mrEF)" has been introduced to better characterize this population.[10] There are several studies suggesting that beta-blockers have an impact on clinical outcomes in patients with mrEF.[11,12]

However, beta-blockers are not a homogeneous class, with distinctions between vasodilating beta-blockers, such as carvedilol and nebivolol, and conventional beta-blockers, such as bisoprolol and metoprolol, based on their vasodilating properties. Vasodilating beta-blockers have favorable effects on central blood pressure, aortic stiffness, and endothelial dysfunction.[13] In addition, vasodilating beta-blockers demonstrate the ability to maintain cardiac index, improve coronary flow reserve, reduce peripheral vascular resistance, improve dyslipidemia, and exert a milder impact on insulin sensitivity.[14,15] Despite these potential advantages, few clinical trials have directly compared the long-term clinical outcomes associated with vasodilating beta-blockers versus conventional beta-blockers. There is a notable lack of research specifically assessing the comparative efficacy of vasodilating beta-blockers versus conventional beta-blockers in patients with AMI and mrEF. This study aims to evaluate the long-term clinical benefits of vasodilating beta-blockers in AMI patients with mrEF in the modern reperfusion era compared to conventional beta-blockers.

## Materials and methods

### Study population and study design

The study population was enrolled from the Korean Acute Myocardial Infarction-National Institutes of Health (KAMIR-NIH) registry.[16] KAMIR-NIH is a nationwide, prospective, multicenter, web-based observational cohort study aiming to develop a prognostic and surveillance index for Korean patients with AMI. Patients who were hospitalized primarily for AMI and signed informed consents were consecutively enrolled from November 2011 to October 2015. Inclusion criteria for the present analysis were consecutive patients aged ≥ 18; ST segment elevation infarction (STEMI) or non-ST segment elevation myocardial infarction (NSTEMI); and patients undergoing PCI. The investigators defined AMI as the criteria for the universal definition of MI.[17] This study protocol was approved by the Institutional Review Board (IRB) at Jeju National University Hospital (IRB No. 2023-08-024) according to the ethical guidelines of the 1975 Declaration of Helsinki. The data for this study were accessed on September 5, 2023, for research purposes. The authors did not have access to any information that could identify individual participants during or after data collection, ensuring strict confidentiality and compliance with ethical standards.

### PCI procedure

Diagnostic coronary angiography and PCI were performed according to current standard procedural guidelines [18] through either the femoral or the radial artery after administration of unfractionated heparin (50–100 IU/kg). Before or during PCI, all patients were given loading doses of 300 mg aspirin and 600 mg loading dose of clopidogrel, 60 mg loading dose of prasugrel or 180 mg loading dose of ticagrelor, unless they had previously received these antiplatelet drugs. During the in-hospital period, the patients received medication, including antiplatelet agents, beta-blockers, renin-angiotensin-aldosterone system (RAAS) inhibitors and lipid lowering agents. After discharge, the patients were encouraged to stay on the same medications they received during hospitalization. The total duration of dual antiplatelet therapy was recommended for more than 12 months to patients who had undergone PCI.

### Definitions and clinical outcomes

The primary outcome was a composite of cardiac death, recurrent MI, or hospitalization for HF during 3-year follow-up period. The secondary outcomes were each component of the primary outcome, all-cause death, any revascularization, stroke, and stent thrombosis during 3-year follow up period. All-cause deaths were attributed to a cardiac cause, unless a definite non-cardiac cause could be established. Recurrent MI was defined as recurrent symptoms with a new electrocardiographic change of ST-segment elevation or re-elevation of cardiac markers to at least twice the upper normal limit after index PCI. Hospitalization for HF was defined as re-hospitalization because of worsening HF requiring more intensive care than continuation of usual treatment in the outpatient department. Any revascularization included repeated PCI or coronary artery bypass surgery on either target or non-target vessels. The clinical follow-ups were routinely performed by visiting the outpatient department of cardiology at 6, 12, 24, and 36 months and whenever any clinical events occurred. If patients did not visit the hospitals, the outcome data were assessed by telephone interview. Clinical events were not centrally adjudicated. The physician identified all events, and the principal investigator of each hospital confirmed them.

### Statistical analysis

The results are expressed as the mean± standard deviation for continuous variables and as counts with percentages for categorical variables. For continuous variables, differences between the two groups were evaluated using the Student's t-test. Categorical variables were analyzed with Pearson's chi-square test between the two groups. Because this study was not randomized, a propensity score-matching (PSM) analysis was performed using the multiple logistic regression model to adjust for any potential confounders. We tested all available variables that could be of potential relevance: age,

sex (male), body mass index (BMI), smoking status (current or ex-smoker), Killip class on admission, LVEF, cardiovascular risk factors (e.g., hypertension, diabetes mellitus, dyslipidemia, HF, chronic kidney disease (CKD), history of MI, history of angina, and history of cerebrovascular disease,), types of MI (STEMI or NSTEMI), coronary reperfusion, left main (LM) or left anterior descending artery (LAD) as infarct-related artery (IRA), multivessel disease (MVD), and comedication (e.g., aspirin, P2Y12 inhibitors, RAAS inhibitors, and statins). Propensity scores were estimated using a multivariable logistic regression model in which treatment group (vasodilating or conventional beta-blocker) was the dependent variable. All baseline variables listed in Table 1 were included as covariates. Patients were then matched 1:1 using nearest neighbor matching without replacement, applying a caliper width of 0.1 standard deviations of the logit of the propensity score. This caliper threshold was chosen to minimize residual bias while preserving sample size, based on established recommendations for optimal matching quality. Various clinical outcomes up to 3-years were estimated by Kaplan-Meier analysis, and differences between the groups were compared with the log-rank test before and after PSM. The Cox proportional hazards model was used to compare the hazard ratio (HR) and 95% confidence intervals (CI) for each clinical outcome in the use of vasodilating beta-blockers and the use of conventional beta-blockers.

**Table 1. Baseline characteristics and medications at discharge.**

| Variables | Entire cohort | | | Propensity score-matched cohort | | |
|---|---|---|---|---|---|---|
| | Vasodilating beta-blockers (n = 1,446) | Conventional beta-blockers (n = 1,216) | P value | Vasodilating beta-blockers (n = 1,054) | Conventional beta-blockers (n = 1,054) | P value |
| Age, years | 63.3 ± 12.5 | 64.5 ± 12.3 | 0.009 | 63.2 ± 12.5 | 63.7 ± 12.2 | 0.415 |
| Male | 1,101 (76.1%) | 873 (71.8%) | 0.011 | 801 (76.0%) | 787 (74.7%) | 0.479 |
| Killip class ≥ II | 234 (16.2%) | 361 (29.7%) | <0.001 | 199 (18.9%) | 230 (21.8%) | 0.094 |
| Body mass index, kg/m$^2$ | 24.1 ± 3.2 | 23.8 ± 3.3 | 0.006 | 24.2 ± 3.2 | 23.9 ± 3.3 | 0.122 |
| Current smoker | 608 (42.0%) | 476 (39.1%) | 0.129 | 434 (41.2%) | 436 (41.4%) | 0.929 |
| Hypertension | 683 (47.2%) | 587 (48.3%) | 0.593 | 491 (50.1%) | 490 (49.9%) | 0.965 |
| Diabetes mellitus | 381 (26.3%) | 370 (30.4%) | 0.020 | 276 (26.2%) | 308 (29.2%) | 0.119 |
| Dyslipidemia | 153 (10.6%) | 109 (9.0%) | 0.163 | 112 (10.6%) | 99 (9.4%) | 0.345 |
| Chronic kidney disease | 238 (16.5%) | 239 (19.7%) | 0.032 | 169 (16.0%) | 178 (16.9%) | 0.597 |
| History of MI | 115 (8.0%) | 111 (9.1%) | 0.278 | 82 (7.8%) | 93 (8.8%) | 0.385 |
| History of angina | 119 (8.2%) | 95 (7.8%) | 0.693 | 80 (7.6%) | 81 (7.7%) | 0.935 |
| History of heart failure | 19 (1.3%) | 15 (1.2%) | 0.854 | 12 (1.1%) | 14 (1.3%) | 0.693 |
| History of stroke | 92 (6.4%) | 94 (7.7%) | 0.168 | 65 (6.2%) | 73 (6.9%) | 0.481 |
| LVEF, % | 45.2 ± 2.8 | 45.3 ± 2.8 | 0.407 | 45.2 ± 2.8 | 45.3 ± 2.7 | 0.427 |
| STEMI | 927 (63.9%) | 744 (61.2%) | 0.149 | 691 (65.6%) | 660 (62.6%) | 0.159 |
| Coronary reperfusion[a] | 1,409 (97.4%) | 1,178 (96.9%) | 0.379 | 1,027 (97.4%) | 1,026 (97.3%) | 0.891 |
| LM or LAD as IRA | 832 (57.5%) | 682 (56.1%) | 0.451 | 615 (58.3%) | 601 (57.0%) | 0.537 |
| MVD | 746 (51.6%) | 645 (53.0%) | 0.455 | 538 (51.0%) | 560 (53.1%) | 0.337 |
| Medications at discharge | | | | | | |
| Aspirin | 1,444 (99.9%) | 1,215 (99.9%) | 0.668 | 1,053 (99.9%) | 1,053 (99.9%) | >0.999 |
| P2Y12 inhibitors | 1,421 (98.3%) | 1,181 (97.1%) | 0.047 | 1,032 (97.9%) | 1,031 (97.8%) | 0.880 |
| RAAS inhibitors | 1,166 (80.6%) | 1,076 (88.5%) | <0.001 | 907 (86.1%) | 919 (87.2%) | 0.443 |
| Statins | 1,386 (95.9%) | 1,135 (93.3%) | 0.004 | 1,008 (95.6%) | 1,000 (94.9%) | 0.412 |

Values are mean ± standard deviation or number (%).

IRA, infarct-related artery; LAD, left anterior descending artery; LM, left main; LVEF, left ventricular ejection fraction; MI, myocardial infarction; MVD, multivessel disease; RAAS, renin-angiotensin-aldosterone system; SD, standardized difference; STEMI, ST-elevation myocardial infarction.

[a] Included reperfusion by percutaneous coronary intervention, thrombolysis, or coronary artery bypass graft, myocardial infarction with non-obstructed coronary arteries, and myocardial bridge.

To determine independent associations between the type of beta-blocker and clinical outcomes, a multivariate Cox regression analysis was performed. The following variables were included in the Cox proportional-hazard regression model as confounding factors; age, sex, smoking status, Killip class on admission, hypertension, diabetes mellitus, dyslipidemia, CKD, LM or LAD as IRA, MVD, LVEF, types of MI, and use of aspirin, P2Y12 inhibitor, RAAS inhibitor, and statin were included as covariates, which were significant on univariable analysis or were generally considered clinically relevant.

For all analyses, a two-sided $P < 0.05$ was regarded as statistically significant. All data were processed with SPSS (version 23.0, SPSS-PC, Inc. Chicago, Illinois) and R version 3.1.3 (R Foundation for Statistical Computing, Vienna, Austria).

## Results

A total of 13,624 consecutive patients were enrolled in the KAMIR-NIH. After excluding 10,962 patients (503 patients who died during the index hospitalization, 921 patients without echocardiographic data, 1,473 patients with LVEF ≤ 40%, 7,626 patients with LVEF ≥ 50%, 416 patients without beta-blockers at discharge, and 23 patients with beta-blockers different from the following (carvedilol, nebivolol, bisoprolol, and metoprolol) at discharge), 2,662 patients were analyzed in this study. After PSM, 1,054 patients in each group were selected (Fig 1).

### Baseline characteristics

In the entire cohort, patients with vasodilating beta-blockers at discharge were more male, and more treated with P2Y12 inhibitors or statins at discharge compared to those with conventional beta-blockers. On the other hand, patients with conventional beta-blockers at discharge were older and had more diabetes mellitus, more CKD, and more treated with RAAS inhibitors at discharge compared to those with vasodilating beta-blockers. However, there was no significant difference

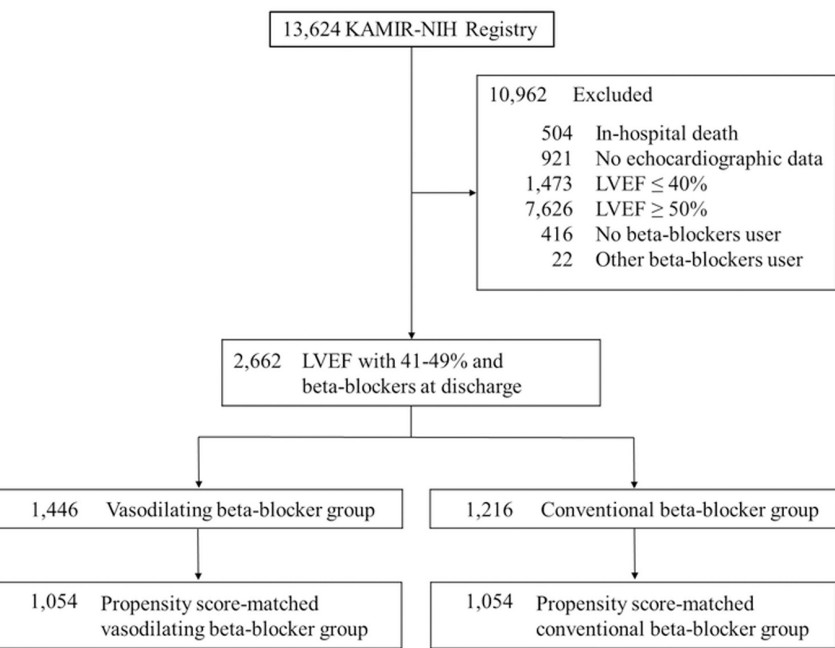

**Fig 1. Flow chart of the group distribution for analysis.** KAMIR-NIH, Korean Acute Myocardial Infarction Registry-National Institute of Health; LVEF, left ventricular ejection fraction.

between the two groups in hypertension, dyslipidemia, history of MI, angina, HF or stroke, LVEF, type of MI, and coronary reperfusion. After PSM, these baseline differences between the two groups were well balanced (Table 1). The mean age of the study patients was 63.5±12.3 years. Of total patients, 1,588 (75.3%) were men, 981 (46.5%) had hypertension, 584 (27.7%) had diabetes mellitus, and 1,351 (64.1%) were STEMI. Overall coronary reperfusion rate was about 97%, and PCI with drug-eluting stents was the main method of coronary reperfusion in the entire cohort and PSM cohort (S1 and S2 Tables).

## Clinical outcomes

The 3-year follow-up rate was 93.7% and 94.0% in the entire and PSM cohorts, respectively. In the entire cohort, 66.5% of patients with vasodilating beta-blocker therapy at discharge continued to take vasodilating beta-blockers at 3-year, only 1.5% had cross-over to conventional beta-blockers. On the other hand, 62.4% of patients with conventional beta-blocker therapy at discharge continued to take conventional beta-blockers, and 6.5% had cross-over to vasodilating beta-blockers. In the PSM cohort, 68.6% of patients with vasodilating beta-blocker therapy at discharge continued to take vasodilating beta-blockers at 3-year, only 1.4% had cross-over to conventional beta-blockers. On the other hand, 63.6% of patients with conventional beta-blocker therapy at discharge continued to take conventional beta-blockers, and 6.0% had cross-over to vasodilating beta-blockers.

In the entire cohort, primary outcomes defined as a composite of cardiac death, MI, or hospitalization for HF occurred in 273 patients (10.3%) at 3-year follow-up. The use of vasodilating beta-blockers at discharge was associated with a lower incidence of primary outcome at 3-year (9.0% versus 11.8%, HR, 0.80; 95% CI, 0.62–0.98; $P=0.039$) than the use of conventional beta-blockers at discharge (Fig 2 and Table 2). However, the incidences of cardiac death (4.1% versus 5.4%, HR, 0.80; 95% CI, 0.55–1.05; $P=0.065$), recurrent MI (2.7% versus 3.0%, HR, 1.00; 95% CI, 0.63–1.48; $P=0.913$), hospitalization for HF (3.5% versus 4.4%, HR, 0.82; 95% CI, 0.55–1.12; $P=0.123$), all-cause death (2.6% versus 3.5%, HR, 0.82; 95% CI, 0.62–1.08; $P=0.154$), any revascularization (8.6% versus 7.7%, HR, 1.14; 95% CI, 0.87–1.50; $P=0.342$), stroke (1.8% versus 2.0%, HR, 0.91; 95% CI, 0.51–1.62; $P=0.749$), and stent thrombosis (0.9% versus 0.8%, HR, 1.15; 95% CI, 0.49–2.68; $P=0.749$) were not significant difference between two groups (S1 Fig and Table 2).

In the PSM cohort, the use of vasodilating beta-blockers at discharge was associated with a lower incidence of primary outcome at 3-year (7.6% versus 11.5%, HR, 0.66; 95% CI, 0.50–0.88; $P=0.004$) than the use of conventional beta-blockers at discharge (Fig 2 and Table 2). Also, the use of vasodilating beta-blocker at discharge was associated with lower incidence of cardiac death (3.3% versus 5.4%, HR, 0.60; 95% CI, 0.39–0.92; $P=0.020$), and all-cause death (5.7% versus 8.4%, HR, 0.67; 95% CI, 0.48–0.93; $P=0.017$) than the use of conventional beta-blockers at discharge. However, the incidences of recurrent MI (1.6% versus 2.8%, HR, 0.62; 95% CI, 0.34–1.14; $P=0.122$), hospitalization for HF (3.2% versus 4.5%, HR, 0.72; 95% CI, 0.46–1.12; $P=0.142$), any revascularization (7.5% versus 7.3%, HR, 1.04; 95% CI, 0.76–1.42; $P=0.821$), stroke (1.6% versus 2.0%, HR, 0.84; 95% CI, 0.44–1.60; $P=0.589$), and stent thrombosis (0.8% versus 0.7%, HR, 1.12; 95% CI, 0.40–3.11; $P=0.833$) were not significant difference between two groups (S2 Fig and Table 2).

Follow-up echocardiography was available in 41% of patients at 1-year, 19% at 2-year, and 18% at 3- year. Among those with follow-up data, the proportion of patients showing a ≥10% improvement in LVEF at 1-year was slightly higher in the vasodilating beta-blocker group compared to the conventional beta-blocker group (13.4% vs. 11.6%; $P=0.104$ in the entire cohort, 14.2% vs. 13.2%; $P=0.331$ in the PSM cohort), though these differences were not statistically significant. Similar trends were observed at 2-year (7.8% vs. 7.2%; $P=0.554$ in the entire cohort, 6.3% vs. 5.8%; $P=0.430$ in the PSM cohort) and at 3-year (5.9% vs 5.6%; $P=0.716$ in the entire cohort, 5.8% vs. 5.1%; $P=0.538$ in the PSM cohort). The proportions of patients who experienced a decline or minimal changes (0–10%) in EF were also comparable between groups. Overall, there were no statistically significant differences in LVEF recovery between vasodilating and conventional beta-blockers in either the entire or PSM cohorts.

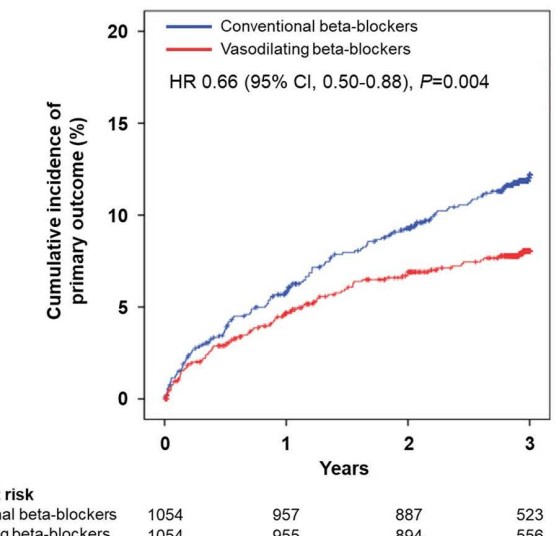

**A) Entire cohort**

**B) Propensity score-matched cohort**

**Number at risk**

| | | | | |
|---|---|---|---|---|
| Conventional beta-blockers | 1216 | 1086 | 1009 | 606 |
| Vasodilating beta-blockers | 1446 | 1299 | 1212 | 726 |

**Number at risk**

| | | | | |
|---|---|---|---|---|
| Conventional beta-blockers | 1054 | 957 | 887 | 523 |
| Vasodilating beta-blockers | 1054 | 955 | 894 | 556 |

**Fig 2. Kaplan-Meier curves and adjusted hazard ratios (HR) for primary outcome during a 3-year follow-up comparing vasodilating beta-blockers versus conventional beta-blockers.**

## Subgroup analysis

We calculated adjusted HR for primary outcomes in various subgroups according to beta-blocker therapy in the PSM cohort. There were no significant interactions between the use of vasodilating beta-blockers and primary outcomes in any of the subgroups (Fig 3). Of note, the beneficial effects of vasodilating beta-blockers were not affected by age, gender, smoking, Killip class, underlying disease, type of MI (STEMI or NSTEMI), LM or LAD as IRA, MVD or statin therapy at discharge.

Carvedilol and bisoprolol were the major beta-blockers, all beta-blockers were used in lower doses than those recommended in the guidelines. (S3 Table)

## Discussion

This study, utilizing a nationwide Korean multicenter registry, demonstrated that the use of vasodilating beta-blockers at discharge were associated with better 3-year primary outcomes such as a composite of cardiac death, recurrent MI, or hospitalization for HF over a 3-year follow-up period in AMI patients with mrEF compared to use of conventional beta-blockers. The present analysis confirmed the long-term clinical benefits of vasodilating beta-blockers, and these positive effects on primary outcome were consistently observed across various subgroups. Additionally, the use of vasodilating beta-blockers was associated with lower incidences of cardiac death, hospitalization for HF, and all-cause mortality compared to the use of conventional beta-blockers, while no significant differences were observed between the two groups for recurrent MI, any revascularization, stroke, or stent thrombosis.

The pivotal role of beta-blockers in AMI management was established in previous studies. The treatment of timolol and metoprolol demonstrated the survival benefits in post-MI patients, [19–21] similarly, the trials evaluating propranolol [22] and atenolol [23] underscored the efficacy of beta-blockers in reducing mortality and recurrent MI. A meta-analysis further supported these findings by showing that long-term beta-blocker treatment for more than 6 months was associated with reduced mortality.[24] These foundational studies set the stage for widespread adoption of beta-blockers in AMI

**Table 2. Clinical outcomes during a 3-year follow-up in patients with mildly reduced left ventricular systolic function according beta-blocker therapy at discharge.**

| Entire cohort | Vasodilating beta-blockers (n = 1,446) | Conventional beta-blockers (n = 1,216) | Adjusted hazard ratio[a] (95% CI) | P value |
|---|---|---|---|---|
| Primary outcome[b] | 130 (9.0) | 143 (11.8) | 0.80 (0.62-0.98) | 0.039 |
| Cardiac death | 59 (4.1) | 66 (5.4) | 0.80 (0.55-1.15) | 0.225 |
| Recurrent myocardial infarction | 39 (2.7) | 36 (3.0) | 1.00 (0.63-1.58) | 0.983 |
| Hospitalization for heart failure | 51 (3.5) | 54 (4.4) | 0.82 (0.55-1.22) | 0.323 |
| All-cause death | 102 (7.1) | 114 (9.4) | 0.82 (0.62-1.08) | 0.154 |
| Any revascularization | 125 (8.6) | 93 (7.7) | 1.14 (0.87-1.50) | 0.342 |
| Stroke | 26 (1.8) | 24 (2.0) | 0.91 (0.51-1.62) | 0.749 |
| Stent thrombosis | 13 (0.9) | 10 (0.8) | 1.15 (0.49-2.68) | 0.749 |
| **Propensity score-matched cohort** | Vasodilating beta-blockers (n = 1,054) | Conventional beta-blockers (n = 1,054) | Adjusted hazard ratio[a] (95% CI) | P value |
| Primary outcome[b] | 80 (7.6) | 121 (11.5) | 0.66 (0.50-0.88) | 0.004 |
| Cardiac death | 35 (3.3) | 57 (5.4) | 0.60 (0.39-0.92) | 0.020 |
| Recurrent myocardial infarction | 17 (1.6) | 29 (2.8) | 0.62 (0.34-1.14) | 0.122 |
| Hospitalization for heart failure | 34 (3.2) | 47 (4.5) | 0.72 (0.46-1.12) | 0.142 |
| All-cause death | 60 (5.7) | 89 (8.4) | 0.67 (0.48-0.93) | 0.017 |
| Any revascularization | 79 (7.5) | 77 (7.3) | 1.04 (0.76-1.42) | 0.821 |
| Stroke | 17 (1.6) | 21 (2.0) | 0.84 (0.44-1.60) | 0.589 |
| Stent thrombosis | 8 (0.8) | 7 (0.7) | 1.12 (0.40-3.11) | 0.833 |

Values are presented as number (%).

CI, confidence interval; HR, hazard ratio; HF, hospitalization for heart failure; MI, myocardial infarction

[a] Adjusted for age, sex, Killip class, hypertension, diabetes mellitus, dyslipidemia, history of myocardial infarction, history of angina, history of heart failure, history of stroke, smoking, chronic kidney disease over stage 3, left main or left anterior descending artery as infarct-related artery, multivessel disease, type of myocardial infarction, and medications (aspirin, P2Y12 inhibitors, renin-angiotensin system blockade, statin).

[b] Defined as a composite cardiac death, recurrent myocardial infarction, or hospitalization for heart failure

management. However, it is essential to recognize that these studies were conducted prior to the modern reperfusion era, with limited use of PCI. In the reperfusion era, subsequent studies have demonstrated that early intravenous beta-blocker administration in patients with AMI reduced the incidence of recurrent MI and ventricular arrhythmias, [25] the CAPRI-CORN (CArvedilol Post-infaRct surVIval COntRolled evaluatioN) trial demonstrated that carvedilol significantly reduced mortality and hospitalization for HF in post-MI patients with LV systolic dysfunction.[5] Although limited evidence exists regarding the efficacy of nebivolol in post-MI patients, prior studies reported that nebivolol was associated with fewer cardiovascular events compared to metoprolol or conventional beta-blockers, with outcomes comparable to those observed in the carvedilol group.[26,27]

Despite these findings, the role of beta-blockers in AMI patients with preserved EF remains uncertain. In the CAPITAL-RCT (Carvedilol Post-Intervention Long-Term Administration in Large-scale Randomized Controlled Trial) trial, which compared carvedilol with no beta-blocker use in STEMI patients with successful PCI and preserved EF, no significant differences were observed between the groups in terms of mortality, MI, or hospitalization for HF.[28] Similarly, the REDUCE-AMI (Randomized Evaluation of Decreased Usage of Beta-Blockers after Acute Myocardial Infarction) trial reported no significant clinical benefits of beta-blockers in AMI patients with LVEF greater than 50%.[29] Conversely, several observational studies have indicated potential benefits, showing that the use of beta-blockers was associated with reduced mortality and MACE in AMI patients with LVEF greater than 40% [30,31] and survival benefits being observed

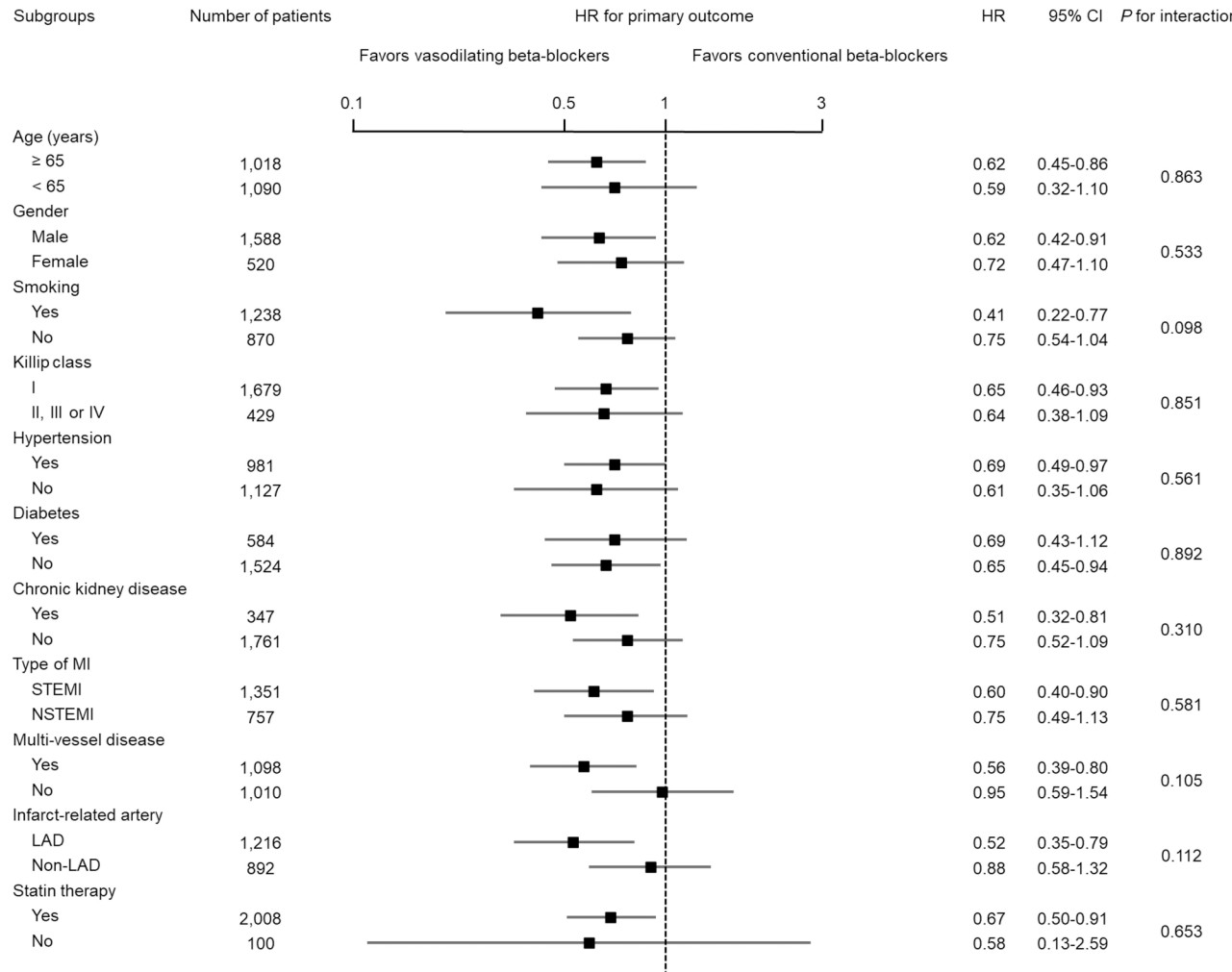

| Subgroups | Number of patients | HR for primary outcome | HR | 95% CI | *P* for interaction |
|---|---|---|---|---|---|
| Age (years) | | | | | |
| ≥ 65 | 1,018 | | 0.62 | 0.45-0.86 | |
| < 65 | 1,090 | | 0.59 | 0.32-1.10 | 0.863 |
| Gender | | | | | |
| Male | 1,588 | | 0.62 | 0.42-0.91 | |
| Female | 520 | | 0.72 | 0.47-1.10 | 0.533 |
| Smoking | | | | | |
| Yes | 1,238 | | 0.41 | 0.22-0.77 | |
| No | 870 | | 0.75 | 0.54-1.04 | 0.098 |
| Killip class | | | | | |
| I | 1,679 | | 0.65 | 0.46-0.93 | |
| II, III or IV | 429 | | 0.64 | 0.38-1.09 | 0.851 |
| Hypertension | | | | | |
| Yes | 981 | | 0.69 | 0.49-0.97 | |
| No | 1,127 | | 0.61 | 0.35-1.06 | 0.561 |
| Diabetes | | | | | |
| Yes | 584 | | 0.69 | 0.43-1.12 | |
| No | 1,524 | | 0.65 | 0.45-0.94 | 0.892 |
| Chronic kidney disease | | | | | |
| Yes | 347 | | 0.51 | 0.32-0.81 | |
| No | 1,761 | | 0.75 | 0.52-1.09 | 0.310 |
| Type of MI | | | | | |
| STEMI | 1,351 | | 0.60 | 0.40-0.90 | |
| NSTEMI | 757 | | 0.75 | 0.49-1.13 | 0.581 |
| Multi-vessel disease | | | | | |
| Yes | 1,098 | | 0.56 | 0.39-0.80 | |
| No | 1,010 | | 0.95 | 0.59-1.54 | 0.105 |
| Infarct-related artery | | | | | |
| LAD | 1,216 | | 0.52 | 0.35-0.79 | |
| Non-LAD | 892 | | 0.88 | 0.58-1.32 | 0.112 |
| Statin therapy | | | | | |
| Yes | 2,008 | | 0.67 | 0.50-0.91 | |
| No | 100 | | 0.58 | 0.13-2.59 | 0.653 |

**Fig 3. Subgroup analysis of the primary outcome in the propensity score-matched cohort comparing vasodilating beta-blockers versus conventional beta-blockers.** CI, confidence interval; LAD, left anterior descending artery; MI, myocardial infarction; STEMI, ST segment elevation myocardial infarction; NSTEMI, non-ST segment elevation myocardial infarction.

even in those with LVEF greater than 50% [32,33] compared to those not receiving beta-blockers. The focus has increasingly shifted to AMI patients with mrEF, a population that appears to exhibit distinct pathophysiological characteristics. Emerging evidence suggests that beta-blockers may confer significant benefits in this subgroup. The use of carvedilol was associated with a lower incidence of the composite outcome, including cardiac death, MI, or hospitalization for HF in STEMI patients with mrEF.[34] Furthermore, the beta-blockers have demonstrated reductions in mortality [11] and MACE [12] in patients with mrEF. These findings suggest that beta-blocker therapy in AMI patients with mrEF may confer substantial benefits reducing cardiac eventsVasodilating beta-blockers, such as carvedilol and nebivolol, have additional vasodilatory properties beyond beta-blockade. Carvedilol has alpha-blocking properties in addition to beta-blocking, which can lead to a reduction in afterload, thereby improving LV function and cardiac output. This can be particularly relevant in patients with HF or reduced EF, where maintaining cardiac output is crucial. Similarly, nebivolol enhances nitric oxide-mediated vasodilation, potentially improving endothelial function and promoting vasodilation.[13–15] These properties

make vasodilating beta-blockers particularly advantageous in AMI patients. The efficacy of vasodilating beta-blockers in AMI patients with preserved and mrEF has been a focus of recent studies. The OBTAIN (Outcomes of Beta-blocker Therapy After Myocardial Infarction) study found no significant difference in mortality between metoprolol and carvedilol in AMI patients with LVEF greater than 40%.[35] In contrast, Chung et al.[27] reported better clinical outcomes with vaso-dilating beta-blockers compared to conventional beta-blockers in AMI patients. Interestingly, subgroup analysis from this study revealed that the benefits of vasodilating beta-blockers were statistically significant in patients with LVEF less than 50%, but no significant differences were observed in those with LVEF greater than 50%. In our study focusing on AMI patients with mrEF, the use of vasodilating beta-blockers demonstrated better clinical outcomes compared to the use of conventional beta-blockers. These findings suggest that AMI patients with mrEF may share certain pathophysiological characteristics with those in the reduced EF category, warranting consideration of similar therapeutic approaches. While the rates of hospitalization due to heart failure (HF) were comparable between vasodilating beta-blockers and conventional beta-blockers, the lower cardiac and all-cause mortality observed with vasodilating beta-blockers suggests potential benefits extending beyond conventional hemodynamic mechanisms such as afterload reduction or ventricular remodeling. Vasodilating beta-blockers exhibit distinct pharmacologic profiles, including antioxidative activity, enhancement of endothelial function, and improved coronary microvascular perfusion.[27,36,37] These additional effects, which are not observed with conventional beta-blockers, may contribute to their favorable impact on survival in patients with AMI and offer a plausible explanation for observed difference in the mortality in our analysis. To validate these observations and refine treatment recommendations, further randomized controlled trials specifically targeting mrEF patients are necessary to establish the optimal beta-blocker therapy for this unique population.

Subgroup analyses suggested that the mortality benefit associated with vasodilating beta-blockers was more pronounced in patients with CKD and in older individuals. In patients with CKD, vasodilating beta-blockers confer hemodynamic benefits, namely a reduction in systemic vascular resistance and enhancement of endothelial function, which may attenuate the deleterious cardiorenal interactions commonly observed in this population. Furthermore, their pleiotropic properties, including antioxidative and anti-inflammatory effects, may provide additional protection in the context of heightened systemic stress.[38] Among older patients, who are more likely to exhibit increased arterial stiffness and vascular dysfunction, the vasodilatory action of these agents may contribute to improved vascular compliance and perfusion.[39] These factors may partly explain the enhanced clinical benefit observed in these subgroups, compared to younger or non-CKD patients.

Several limitations of this study should be acknowledged. As this analysis was based on non-randomized, observational registry data, the potential for inherent bias remains. The decision to prescribe beta-blockers was left to the discretion of individual physicians, and the registry did not collect information on the reasons for non-prescription in some patients at discharge. Although PSM was employed to adjust for baseline differences, unmeasured and residual confounding factors, as well as possible selection bias, may have persisted. Notably, renal function was stratified using CKD stages rather than estimated glomerular filtration rate, which may have limited the granularity of renal risk assessment. Treatment selection between vasodilating and conventional beta-blockers after PCI was also based on physician preference, which may have introduced allocation bias beyond the control of statistical matching. Additionally, the study did not capture detailed information on beta-blocker dosing, therapy modifications during follow-up, adverse events, or direct measures of medication adherence. Adherence was inferred from prescription continuity at follow-up visits and appeared similar between treatment groups over 1- and 2-year intervals; however, this indirect measure does not confirm actual medication intake. Furthermore, real-time adherence behavior, dose titrations, and interim changes in therapy were not systematically recorded. The registry also did not systematically record the use of important HF-related medications sodium glucose cotransporter 2 inhibitor, mineralocorticoid receptor antagonists and angiotensin receptor neprilysin inhibitor. This limitation reflects the AMI-focused scope of the KAMIR-NIH registry, which was not specifically designed to capture the full spectrum of HF pharmacotherapy. These gaps raise the possibility of therapeutic confounding over the

course of follow-up. Moreover, conventional beta-blockers were generally prescribed at sub-target doses relative to those recommended in HF clinical trials. Although this likely reflects real-world prescribing behavior in AMI settings, suboptimal dosing may have attenuated their observed effectiveness. As the dose information was only available at discharge and follow-up titration was not systematically documented, we were unable to incorporate dose intensity into the PSM model. Thus, variation in dosing across groups represents a further source of potential bias. LVEF was determined through echocardiography, with mrEF classification based exclusively on these imaging results. Echocardiographic assessments were conducted independently at each participating institution according to routine clinical protocols, without subsequent adjudication by a centralized core laboratory. Consequently, potential variability in measurement across centers may have affected the precision of LVEF categorization, particularly in cases near the mrEF threshold, where minor deviations could influence group assignment. Due to registry constraints, detailed coronary anatomical data-such as bifurcation lesions and stenting strategies-were unavailable, and validated angiographic risk scores like the SYNTAX score could not be applied. Although the presence of LM disease was documented and infrequent across both cohorts (1.7%), the absence of detailed anatomical characterization limits our ability to assess lesion complexity as a factor influencing outcomes. The clinical events were not centrally adjudicated in our multicenter national prospective registry. Instead, events were identified by the treating physicians and subsequently confirmed by the principal investigator of each hospital. This decentralized approach may have led to variability in data collection and the potential omission of certain clinical events in the database. The predominant use of carvedilol and bisoprolol in this study may limit the generalizability of the results, as the utilization of nebivolol and metoprolol is uncommon in Korea. Caution should be exercised when extrapolating these findings to countries with different prescribing patterns of beta-blockers. Finally, the results of this study cannot be applied to patients of other racial and ethnic groups because the population of this study consisted of a single ethnicity of Korean patients. To ascertain the impact of race and ethnicity on the factors studied, further investigations involving diverse patient populations are warranted.

In conclusion, the use of vasodilating beta-blockers at discharge in patients with AMI and mildly reduced LV systolic function who survived the initial attack was associated with better long-term clinical outcomes compared with the use of conventional beta-blockers. These findings suggest that vasodilating beta-blockers may be considered as a therapeutic option to improve clinical outcomes in this patient population.

## Supporting information

**S1 Fig. Kaplan-Meier curves and adjusted hazard ratios (HR) for 3-year clinical events in the entire cohort with vasodilating beta-blockers vs. conventional beta-blockers.** (A) Cardiac death (B) Recurrent myocardial infarction. (C) Hospitalization for heart failure. (D) All-cause death. (E) Any revascularization. (F) Stroke. (G) Stent thrombosis. CI, confidence interval.
(TIF)

**S2 Fig. Kaplan-Meier curves and adjusted hazard ratios (HR) for 3-year clinical events in the propensity score-matched cohort with vasodilating beta-blockers vs. conventional beta-blockers.** (A) Cardiac death (B) Recurrent myocardial infarction. (C) Hospitalization for heart failure. (D) All-cause death. (E) Any revascularization. (F) Stroke. (G) Stent thrombosis. CI, confidence interval.
(TIF)

**S1 Table. Reperfusion rates and methods in the entire cohort.**
(PDF)

**S2 Table. Reperfusion rates and methods in the propensity-score matched cohort.**
(PDF)

**S3 Table. Generic names and doses of beta-blockers at discharge in the propensity-score matched cohort.** (PDF)

## Acknowledgments

We appreciate the contribution of the KAMIR-NIH investigators: Myung Ho Jeong, MD (The lead investigator), Chonnam National University Hospital, Gwangju, Republic of Korea, Tae Hoon Ahn, MD, Department of Cardiology, Gil Medical Center, Gachon University College of Medicine, Incheon, Republic of Korea, Ki-Bae Seung, MD, Cardiology Division, Department of Internal Medicine, College of Medicine, The Catholic University of Korea, Seoul, Republic of Korea, Chong-Jin Kim, MD, Kyunghee University Hospital at Gangdong, Seoul, Republic of Korea, Shung Chull Chae, MD, Department of Internal Medicine, Kyungpook National University Hospital, Daegu, Republic of Korea, Jin-Yong Hwang, MD, Department of Internal Medicine, Gyeonsang National University School of Medicine, Gyeongsang National University Hospital, Jinju, Republic of Korea, Seung-Ho Hur, MD, Keimyung University Dongsan Medical Center, Cardiovascular Medicine, Daegu, Republic of Korea, Seung-Woon Rha, MD, Cardiovascular Center, Korea University Guro Hospital, Seoul, Republic of Korea, Kwang Soo Cha, MD, Pusan National University Hospital, Busan, Republic of Korea, Chang-Hwan Yoon, MD, Cardiovascular Center, Seoul National University Bundang Hospital, Seongnam, Republic of Korea, Hyo-Soo Kim, MD, Cardiovascular Center, Department of Internal Medicine, Seoul, Republic of Korea, Hyeon-Cheol Gwon, MD, Heart Vascular and Stroke Institute, Samsung Medical Center, Sungkyunkwan University School of Medicine, Seoul, Republic of Korea, Jung-Hee Lee, MD, Division of Cardiology, Yeungnam University Medical Center, Yeungnam University College of Medicine, Daegu, Republic of Korea, Seok Kyu Oh, MD, Division of Cardiology, Department of Internal Medicine, Wonkwang University School of Medicine, Iksan, Republic of Korea, Junghan Yoon, MD, Division of Cardiology, Department of Internal Medicine, Yonsei University Wonju College of Medicine, Wonju Severance Christian Hospital, Wonju, Republic of Korea, Jei Keon Chae, MD, Division of Cardiology, Department of Internal Medicine, Chonbuk National University Medical School, Jeonju, Republic of Korea, In-Whan Seong, MD, Department of Internal Medicine, Chungnam National University Hospital, Chungnam National University College of Medicine, Daejeon, Republic of Korea, Kyung-Kuk Hwang, MD, Department of Internal Medicine, Chungbuk National University College of Medicine, Chungbuk Regional Cardiovascular Center, Division of Cardiology, Department of Internal Medicine, Chungbuk National University Hospital, Cheongju, Republic of Korea, Doo-Il Kim, MD, Department of Internal Medicine, Inje University College of Medicine, Haeundae Paik hospital, Busan, Republic of Korea.

## Author contributions

**Conceptualization:** Ki Yung Boo, Miyeon Kim, Jae-Geun Lee, Myung Ho Jeong.

**Data curation:** Ki Yung Boo, Miyeon Kim, Geum Ko, Joon Hyouk Choi, Seung-Ho Hur, Kwang Soo Cha, Myung Ho Jeong.

**Formal analysis:** Ki Yung Boo, Miyeon Kim, Geum Ko, Joon Hyouk Choi.

**Funding acquisition:** Jae-Geun Lee.

**Investigation:** Jae-Geun Lee, Joon Hyouk Choi.

**Methodology:** Jae-Geun Lee, Song-Yi Kim, Jin-Yong Hwang, Seung-Ho Hur, Kwang Soo Cha, Myung Ho Jeong.

**Resources:** Jin-Yong Hwang, Seung-Ho Hur, Kwang Soo Cha, Myung Ho Jeong.

**Supervision:** Jae-Geun Lee, Seung-Jae Joo.

**Validation:** Jae-Geun Lee.

**Visualization:** Song-Yi Kim.

**Writing – original draft:** Ki Yung Boo, Miyeon Kim.

**Writing – review & editing:** Jae-Geun Lee, Seung-Jae Joo.

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
