## [Decision Letter · Decision Letter 0]

Dear Dr. Lee,

We look forward to receiving your revised manuscript.

Kind regards,

Ahmed Qasim Mohammed Alhatemi, Mbchb

Academic Editor

PLOS ONE

Journal Requirements:

3. Thank you for uploading your study's underlying data set. Unfortunately, the repository you have noted in your Data Availability statement does not qualify as an acceptable data repository according to PLOS's standards.

4. One of the noted authors is a group or consortium [KAMIR-NIH registry investigators]. In addition to naming the author group, please list the individual authors and affiliations within this group in the acknowledgments section of your manuscript. Please also indicate clearly a lead author for this group along with a contact email address.

Reviewers' comments:

Reviewer's Responses to Questions

**Comments to the Author**

1. Is the manuscript technically sound, and do the data support the conclusions?

Reviewer #1: Yes

Reviewer #2: Yes

Reviewer #3: Yes

Reviewer #4: Yes

2. Has the statistical analysis been performed appropriately and rigorously?

Reviewer #1: Yes

Reviewer #2: Yes

Reviewer #3: Yes

Reviewer #4: Yes

3. Have the authors made all data underlying the findings in their manuscript fully available?

Reviewer #1: Yes

Reviewer #2: Yes

Reviewer #3: Yes

Reviewer #4: Yes

4. Is the manuscript presented in an intelligible fashion and written in standard English?

Reviewer #1: Yes

Reviewer #2: No

Reviewer #3: Yes

Reviewer #4: Yes

Reviewer #1: Strengths of the Study:

- Relevant Research Question: The role of vasodilating beta-blockers in AMI with mildly reduced EF (40-49%) is clinically debated, making this study highly relevant.

- Robust Dataset: The use of a targeted patient subset from the large KAMIR-NIH registry strengthens external validity.

- Well-Defined Clinical Outcomes: The composite endpoint (cardiac death, recurrent MI, HF hospitalization) is clinically meaningful.

- Appropriate Use of Propensity Score Matching: Helps reduce selection bias, making the groups more comparable.

- Findings Suggest Clinical Impact: Vasodilating beta-blockers appear to be associated with better long-term outcomes, particularly in reducing cardiac death and HF hospitalization.

Minor Concerns and Suggested Improvements:

1. Residual Confounding

While PSM is a valid approach, unmeasured confounders (e.g., medication adherence, renal function, physician preference) may still exist.

Suggestion: The authors may consider acknowledging this as a limitation in the discussion.

2. Vasodilating Beta-Blocker Subgroup Analysis

The study groups all vasodilating beta-blockers (Carvedilol, Nebivolol) together despite different mechanisms.

However, since the authors report no significant difference between these agents, separate subgroup analysis is not necessary.

Suggestion: Briefly discuss the availability of Nebivolol in post-MI treatment guidelines (as some countries prefer Carvedilol due to stronger RCT evidence).

3. Medication Adherence & Dose Titration

Was medication adherence assessed? If patients on vasodilating beta-blockers were more adherent, this could bias results.

Suggestion: Acknowledge this limitation in the discussion.

- Final Recommendation: Minor revisions. The manuscript does not require major revisions but would benefit from clarifying residual confounding, sensitivity analysis, and minor language refinements.

Conclusion

Overall, this study provides valuable insights into the long-term benefits of vasodilating beta-blockers in AMI patients with mildly reduced EF. While the methodology is robust, minor improvements in addressing potential residual confounding and medication adherence would strengthen its conclusions. Given the clinical relevance of this topic and the study’s well-executed design, I recommend acceptance with minor revisions.

Reviewer #2: The authors present the results of a PSM analysis suggesting that vasodilating betablockers might have some benefit in patients with AMI and LVEF between 41 – 49% when compared to non-vasodilating BB (mainly Carvedilol vs bisoprolol). This is an interesting analysis considering the most recent evidence showing the absence of benefit of BB in patients with AMI and LVEF >50%, as it explores a subgroup of patients where evidence is lacking. Also, it brings some data in the current era of adequate reperfusion strategies for patients with AMI.

Some comments.

1. The authors need to specify how was LVEF obtained? Echo only? CMR?, do they have inter or intravariability analysis? Did LVEF was assessed in some core lab? This is relevant since the range between 41-49% is small and subjected to variability. This is also a problem with the current HFmrEF classification for patients with HF.

2. Do they have any data on the use of other medications? SGLT2i, MRA, ARNI?

3. Do they have data on LVEF recovery? How many patients developed heart failure? And how many showed a recovery? Is there any difference between vasodilating and non-vasodilating betablockers?

4. The authors state that there is a reduction of heart failure hospitalizations, but this seems to be non-significant.

5. The reduction in the primary outcome seems to be driven by Cardiac death and all cause death but no HFH, so there should be another explanation for the possible benefit different from afterload reduction and remodeling…

Reviewer #3: This is an interesting analysis in which 1054 patients with vasodilating BBs were propensity-score matched with 1054 patients using conventional BB groups among patients with midrange LVEF (41-49%) and beta-blockers at discharge.

Authors report impressie 20% risk reduction with the use of vasodilating BBs in terms of cumulative incidence of the primary outcome in the entire cohort and even 34% when groups are propensity matched.

Authors provide a myriad of supplementary tables and graphs.

The analysis is intricate, however, there are some major shortcomings.

Pharmacotherapy adjustement during follow-up migh be inadequate, we do not know which patients were using MRAs, loop diuretics, etc. and in which doses. I am aware that authors adjusted their analysis for P2Y12 inhibitors, RAAS inhibitors and statins, however we do not know what occured with other relevant medications in this population, although many parameters were taken into account regarding PSM co-variate adjustment.

All patients had ischemia-induced systolic dysfunction since these were all AMI patients with mild systolic dysfunction.

How do the authors explain that vasodilating BBs had stronger impact in patients with chronic kidney disease versus ones that did not have it? What is the biological rationale and plausibility behind this? Similarly, why did the patients that were older had more benefit from it, compared to younger ones?

There is also lack of characterizing severity burden of coronary disease which might have significantly impact on the outcomes that authors report. How many had bifurcation lesions and had bifurcation stenting? How many had LM disease? Authors also do not provide any objective parameter for this such as for example SYNTAX score or similar. This is a limitation to this study.

Finally, mean doses of adminstered beta-blockers seem rather low in the conventional BB groups. Just a 1.9 mg of bisoprolol on average and 58 mg of metoprolol. Could authors adjust their analysis adjusted for the dosages. We know that this might be a considerable limitations since we know that there is incremental survival and other gains increasing with higher doses of BBs administered, in HF in particular. If these patients were underdosed then we cannot expect beneficial effects of these drugs as tested in pivotal randomized controlled trials. This critical fact needs to be addressed.

Reviewer #4: In the manuscript entitled “Vasodilating beta blocker in AMI patients with miildly reduced LV function” the authors used a nationwide MI database in South Korea to determine if the type of beta blocker used in treatment of patients with LV EF 40-50%, affects cardiac death, recurrent MI or hospitalization for heart failure over 3 years. The group was divided based on whether subjects were treated with vasodilating beta blocker or conventional beta blocker. The analysis appears relatively well executed and approach clearly described.

Questions:

1) the authors use a propensity matching approach to compare the effect of vasodilating vs conventional beta blocker to assess the effects of different beta blocker treatment on patients with similar clinical characteristics. On page 6 the authors indicate the matching strategy was made using a propensity score with nearest neighbor matching algorithm, matched with a caliper width equal to 0.1 of the SD of the propensity score. It would be useful if the authors could clarify how the propensity score is derived, and what exactly the caliper width refers to.

2) Is there any relationship in the data between LV EF (40-50%) and event rates? Do vasodilating beta blockers have greater effect in lower LV EF or no clear effect?

**Do you want your identity to be public for this peer review?** For information about this choice, including consent withdrawal, please see our Privacy Policy

Reviewer #1: **Yes: ** Ibrahim A M Abdulhabeeb

Reviewer #2: **Yes: ** Edgar Francisco Carrizales Sepulveda

Reviewer #3: **Yes: ** Josip Andelo Borovac, MD, PHD, FESC

Reviewer #4: No

---

## [Author Response · Author response to Decision Letter 1]

8 May 2025

Reviewer #1

1. Residual Confounding

While PSM is a valid approach, unmeasured confounders (e.g., medication adherence, renal function, physician preference) may still exist.

Suggestion: The authors may consider acknowledging this as a limitation in the discussion.

- Thank you very much for bringing this to our attention.

- We would like to clarify that medication adherence was already acknowledged as a limitation in the original manuscript; however, we have revised and clarified this point further in the revised version. Additionally, we have included a new statement addressing the limitation related to renal function, emphasizing that it was classified by CKD stage rather than estimated GFR, which may have limited the precision of renal risk stratification. Regarding physician treatment preference, this was already acknowledged as a limitation in the initial version, and thus, no additional changes were made in this respect.

- We added the following to the limitation in the discussion section:

Although PSM was employed to adjust for baseline differences, unmeasured and residual confounding factors, as well as possible selection bias, may have persisted. Notably, renal function was stratified using CKD stages rather than estimated glomerular filtration rate, which may have limited the granularity of renal risk assessment. Treatment selection between vasodilating and conventional beta-blockers after PCI was also based on physician preference, which may have introduced allocation bias beyond the control of statistical matching.

2. Vasodilating Beta-Blocker Subgroup Analysis

The study groups all vasodilating beta-blockers (Carvedilol, Nebivolol) together despite different mechanisms. However, since the authors report no significant difference between these agents, separate subgroup analysis is not necessary.

Suggestion: Briefly discuss the availability of Nebivolol in post-MI treatment guidelines (as some countries prefer Carvedilol due to stronger RCT evidence).

- Thank you for your comment.

- We analyzed clinical outcomes by dividing patients into carvedilol group and nebivolol group, and found no significant differences between the two groups in terms of primary outcome (7.7% vs 6.5%, HR 1.29, 95% CI: 0.58-2.86, P=0.531), cardiac death (3.3% vs 3.7%, HR 0.73, 95% CI: 0.25-2.14, P=0.568 ), MI (1.5% vs 2.8%, HR 0.82, 95% CI: 0.22-3.01, P=0.759), hospitalization due to HF (3.6% vs 0.9%, HR 6.05, 95% CI: 0.83-44.14, P=0.076), all-cause death (5.6% vs 6.5%, HR 0.76, 95% CI: 0.34-1.70, P=0.500), any revascularization (7.6% vs 6.5%, HR 1.43, 95% CI: 0.65-3.16, P=0.376), stroke (1.5% vs 2.8%, HR 0.52, 95% CI: 0.15-1.84, P=0.307), and stent thrombosis (0.8% vs 1.8%, HR 0.47, 95% CI: 0.23-1.76, P=0.106).

- We added the following to the Discussion section:

Although limited evidence exists regarding the efficacy of nebivolol in post-MI patients, prior studies reported that nebivolol was associated with fewer cardiovascular events compared to metoprolol or conventional beta-blockers, with outcomes comparable to those observed in the carvedilol group. (26,27)

Reference)

26. Ozaydin M, Yucel H, Kocyigit S, Adali MK, Aksoy F, Kahraman F, et al. Nebivolol versus Carvedilol or Metoprolol in Patients Presenting with Acute Myocardial Infarction Complicated by Left Ventricular Dysfunction. Med Princ Pract. 2016;25(4):316-22.

27. Chung J, Han JK, Yang HM, Park KW, Kang HJ, Koo BK, et al. Long-term efficacy of vasodilating beta-blocker in patients with acute myocardial infarction: nationwide multicenter prospective registry. Korean J Intern Med. 2021;36(Suppl 1):S62-S71. 

3. Medication Adherence Dose Titration

Was medication adherence assessed? If patients on vasodilating beta-blockers were more adherent, this could bias results.

Suggestion: Acknowledge this limitation in the discussion.

- We appreciate your valuable comment.

- We assessed the adherence to medication. Medication adherence was indirectly assessed based on prescription data at follow-up. In the entire cohort, 97% of patients discharged on vasodilating beta-blockers remained on the same regimen at 1 year, compared to 95% in patients with conventional beta-blockers. At 2 years, persistence rates were identical between the groups, with 73% of patients with both the vasodilating and conventional beta-blockers continuing their initially prescribed therapy. In the PSM cohort, medication persistence showed a comparable trend: 97% and 74% of patients with vasodilating beta-blockers remained on therapy at 1 and 2 years, respectively, compared with 96% and 74% of patients with conventional beta-blockers. There are no significant differences between vasodilating beta-blockers and conventional beta-blockers.

- We added the following to the limitation in the discussion section:

Additionally, the study did not capture detailed information on beta-blocker dosing, therapy modifications during follow-up, adverse events, or direct measures of medication adherence. Adherence was inferred from prescription continuity at follow-up visits and appeared similar between treatment groups over 1- and 2-year intervals; however, this indirect measure does not confirm actual medication intake. Furthermore, real-time adherence behavior, dose titrations, and interim changes in therapy were not systematically recorded. 

Reviewer #2:

1. The authors need to specify how was LVEF obtained? Echo only? CMR?, do they have inter or intravariability analysis? Did LVEF was assessed in some core lab? This is relevant since the range between 41-49% is small and subjected to variability. This is also a problem with the current HFmrEF classification for patients with HF.

- Thank you for your comment.

- We added the following to the limitation in the discussion section:

LVEF was determined through echocardiography, with mrEF classification based exclusively on these imaging results. Echocardiographic assessments were conducted independently at each participating institution according to routine clinical protocols, without subsequent adjudication by a centralized core laboratory. Consequently, potential variability in measurement across centers may have affected the precision of LVEF categorization, particularly in cases near the mrEF threshold, where minor deviations could influence group assignment.

2. Do they have any data on the use of other medications? SGLT2i, MRA, ARNI?

- Thank you very much for bringing this to our attention.

- The KAMIR-NIH registry does not include detailed information on the use of certain heart failure-specific medications such as SGLT2 inhibitors, MRAs, or ARNIs. As the registry was primarily designed to investigate acute myocardial infarction rather than heart failure management, comprehensive data on all heart failure therapies were not systematically collected. Furthermore, the enrollment period of the registry (2011–2015) preceded the widespread adoption of SGLT2 inhibitors and ARNIs in Korean clinical practice, which may further explain the absence of these variables.

- We added the following to the limitation in the discussion section:

The registry also did not systematically record the use of important heart failure-related medications such as sodium glucose cotransporter 2 inhibitor, mineralocorticoid receptor antagonists and angiotensin receptor neprilysin inhibitor. This limitation reflects the AMI-focused scope of the KAMIR-NIH registry, which was not specifically designed to capture the full spectrum of heart failure pharmacotherapy. These gaps raise the possibility of therapeutic confounding over the course of follow-up.

3. Do they have data on LVEF recovery? How many patients developed heart failure? And how many showed a recovery? Is there any difference between vasodilating and non-vasodilating betablockers?

- We appreciate the reviewer’s interest in the longitudinal changes in LVEF.

- Follow-up echocardiography was not universally performed, as it was not mandated by the study protocol and was conducted at the discretion of the treating physician. In the entire cohort, follow-up echocardiography was available in 41% of patients at 1 year, 19% at 2 years, and 18% at 3 years. Among these, the proportions of patients showing a ≥10% improvement in LVEF were slightly higher in the vasodilating beta-blocker group compared to the conventional group, but these differences were not statistically significant at any time point. Specifically, at 1 year, the rate of significant LVEF improvement (≥10%) was 13.4% in the vasodilating beta-blocker group and 11.6% in the conventional group; at 2 years, 7.8% vs. 7.2%; and at 3 years, 5.9% vs. 5.6%, respectively. A similar trend was observed in the PSM cohort, with 14.2% vs. 13.2% at 1 year, 6.3% vs. 5.8% at 2 years, and 5.8% vs. 5.1% at 3 years for vasodilating and conventional beta-blockers, respectively. The proportions of patients who experienced a decline in EF or minimal changes (0–10%) were also comparable between groups. Overall, there were no statistically significant differences in LVEF recovery between vasodilating and conventional beta-blockers in either the entire or PSM cohorts.

4. The authors state that there is a reduction of heart failure hospitalizations, but this seems to be non-significant.

- Thank you very much for your comments.

- We agree with the reviewer’s comments. Although the incidence of hospitalization for heart failure was numerically lower in the vasodilating beta-blocker group, the difference did not reach statistical significance in our analysis. This has been presented in Table 2 as the results of both the entire cohort and the PSM cohort.

5. The reduction in the primary outcome seems to be driven by Cardiac death and all cause death but no HFH, so there should be another explanation for the possible benefit different from afterload reduction and remodeling…

- Thank you for the great comments.

- We added the following in the Discussion section:

- While the rates of hospitalization due to heart failure were comparable between vasodilating beta-blockers and conventional beta-blockers, the lower cardiac and all-cause mortality observed with vasodilating beta-blockers suggests potential benefits extending beyond conventional hemodynamic mechanisms such as afterload reduction or ventricular remodeling. Vasodilating beta-blockers exhibit distinct pharmacologic profiles, including antioxidative activity, enhancement of endothelial function, and improved coronary microvascular perfusion. (27, 36, 37) These additional effects, which are not observed with conventional beta-blockers, may contribute to their favorable impact on survival in patients with AMI and offer a plausible explanation for observed difference in the mortality in our analysis.

References)

27 .Chung J, Han JK, Yang HM, Park KW, Kang HJ, Koo BK, et al. Long-term efficacy of vasodilating beta-blocker in patients with acute myocardial infarction: nationwide multicenter prospective registry. Korean J Intern Med. 2021;36(Suppl 1):S62-S71.

36. DiNicolantonio JJ, Lavie CJ, Fares H, Menezes AR, O'Keefe JH. Meta-analysis of carvedilol versus beta 1 selective beta-blockers (atenolol, bisoprolol, metoprolol, and nebivolol). Am J Cardiol. 2013;111(5):765-9.

37. Sorrentino SA, Doerries C, Manes C, Speer T, Dessy C, Lobysheva I, et al. Nebivolol exerts beneficial effects on endothelial function, early endothelial progenitor cells, myocardial neovascularization, and left ventricular dysfunction early after myocardial infarction beyond conventional beta1-blockade. J Am Coll Cardiol. 2011;57(5):601-11.

Reviewer #3:

1. Pharmacotherapy adjustement during follow-up migh be inadequate, we do not know which patients were using MRAs, loop diuretics, etc. and in which doses. I am aware that authors adjusted their analysis for P2Y12 inhibitors, RAAS inhibitors and statins, however we do not know what occured with other relevant medications in this population, although many parameters were taken into account regarding PSM co-variate adjustment.

- Thank you for the great comments. We agree with the reviewer’s valuable comment.

- Unfortunately, information on the use and dosing of certain heart failure-specific medications such as MRAs or loop diuretics was not collected in the KAMIR-NIH registry. This is likely because the registry was primarily designed to evaluate AMI rather than chronic heart failure management. In addition, medication adherence was assessed only at predefined outpatient follow-up visits based on prescription data, and actual medication intake or dose titration could not be monitored.

- We added the following to the limitation in the discussion section:

The registry also did not systematically record the use of important heart failure-related medications such as sodium glucose cotransporter 2 inhibitor, mineralocorticoid receptor antagonists and angiotensin receptor neprilysin inhibitor. This limitation reflects the AMI-focused scope of the KAMIR-NIH registry, which was not specifically designed to capture the full spectrum of heart failure pharmacotherapy. These gaps raise the possibility of therapeutic confounding over the course of follow-up.

-

2. How do the authors explain that vasodilating BBs had stronger impact in patients with chronic kidney disease versus ones that did not have it? What is the biological rationale and plausibility behind this? Similarly, why did the patients that were older had more benefit from it, compared to younger ones?

- Thank you for the great comments.

- We appreciate the reviewer’s insightful comment. The more pronounced benefit of vasodilating beta-blockers in patients with CKD may be attributable to their favorable hemodynamic and endothelial effects, which are particularly relevant in this population. Vasodilating beta-blockers reduces peripheral vascular resistance and improve endothelial function - mechanisms that may help mitigate the progression of cardiorenal syndrome and reduce cardiovascular stress in patients with impaired renal function. Furthermore, their antioxidative and anti-inflammatory properties may offer additional protective effects in CKD, where oxidative stress and systemic inflammation are commonly elevated.

Similarly, the greater benefit observed in older patients could be explained by the age-related decline in vascular compliance and increased arterial stiffness, both of which may be better addressed by the vasodilatory properties of these agents. In this context, vasodilating beta-blockers may offer more comprehensive cardiovascular protection than conventional beta-blockers, particularly in high-risk populations such as older adults and those with CKD. We have included this possible biological rationale in the revised discussion.

- Subgroup analyses suggested that the mortality benefit associated with vasodilating beta-blockers was more pronounced in patients with CKD and in older individuals. In patients with CKD, vasodilating beta-blockers confer hemodynamic benefits, namely a reduction in systemic vascular resistance and enhancement of endothelial function, which may attenuate the deleterious cardiorenal interactions commonly observed in this population. Furthermore, their pleiotropic properties, including antioxidative and anti-inflammatory effects, may provide additional protection in the context of heightened systemic stress.(38) Among older patients, who are more likely to exhibit increased arterial stiffness and vascular dysfunction, the vasodilatory action of these agents may contribute to improved vascular compliance and perfusion.(39) These factors may partly explain the enhanced clinical benefit observed in these subgroups, compared to younger or non-CKD patients.

References)

38. Wali RK, Iyengar M, Beck GJ, Chartyan DM, Chonchol M, Lukas MA, et al. Efficacy and safety of carvedilol in treatment of heart failure with chronic kidney disease: a meta-analysis of randomized trials. Circ Heart Fail. 2011;4(1):18-26.

39. Patel K, Fonarow GC, Ekundayo OJ, Aban IB, Kilgore ML, Love TE, et al. Beta-blockers in older patients with heart failure and preserved ejection fraction: class, dosage, and outcomes. Int J Cardiol. 2014;173(3):393-401. 

3. There is also lack of characterizing severity burden of coronary disease which mi

---

## [Decision Letter · Decision Letter 1]

Dear Dr. Lee,

We look forward to receiving your revised manuscript.

Kind regards,

Ahmed Qasim Mohammed Alhatemi, Mbchb

Academic Editor

PLOS ONE

Journal Requirements:

Reviewers' comments:

Reviewer's Responses to Questions

**Comments to the Author**

Reviewer #1: All comments have been addressed

Reviewer #2: (No Response)

Reviewer #3: All comments have been addressed

Reviewer #4: All comments have been addressed

2. Is the manuscript technically sound, and do the data support the conclusions?

Reviewer #1: Yes

Reviewer #2: Yes

Reviewer #3: Yes

Reviewer #4: Yes

3. Has the statistical analysis been performed appropriately and rigorously?

Reviewer #1: Yes

Reviewer #2: Yes

Reviewer #3: N/A

Reviewer #4: Yes

4. Have the authors made all data underlying the findings in their manuscript fully available?

Reviewer #1: Yes

Reviewer #2: Yes

Reviewer #3: Yes

Reviewer #4: Yes

5. Is the manuscript presented in an intelligible fashion and written in standard English?

Reviewer #1: Yes

Reviewer #2: Yes

Reviewer #3: Yes

Reviewer #4: Yes

Reviewer #1: The revised manuscript has successfully addressed the previous reviewer comments. The study is technically sound and well-conducted using a large national registry (KAMIR-NIH), and the propensity score-matched analysis is appropriate. The clarification of limitations (e.g., residual confounding, medication adherence, subgroup rationale) enhances the transparency and integrity of the findings. The authors provided clear explanations regarding the observed benefits of vasodilating beta-blockers in subgroups such as CKD and elderly patients. The clinical message is relevant, especially in the context of real-world use of beta-blockers post-AMI in patients with mildly reduced LVEF. I recommend acceptance of this version.

Reviewer #2: Thanks to the authors for the great effort addressing our comments.

Please try to include the data used in your responses as a part of the results. Example, you showed data on LVEF longitudinal changes as a part of your responses to the reviewers, but that data is not in the manuscript, and seems to be valuable.

Thanks

Reviewer #3: Thank you for answering my queries. No further questions. While the paper still has some limits, most of the important issues have been dealt with.

Reviewer #4: The authors have adequately addressed the queries from reviewers and taken care to modify the text of the article to reflect areas of uncertainty raised by reviewers.

**Do you want your identity to be public for this peer review?** For information about this choice, including consent withdrawal, please see our Privacy Policy

Reviewer #1: **Yes: ** Ibrahim A M Abdulhabeeb, MD

Reviewer #2: **Yes: ** Edgar Francisco Carrizales-Sepúlveda

Reviewer #3: No

Reviewer #4: No

---

## [Author Response · Author response to Decision Letter 2]

29 May 2025

Reviewer #1:

The revised manuscript has successfully addressed the previous reviewer comments. The study is technically sound and well-conducted using a large national registry (KAMIR-NIH), and the propensity score-matched analysis is appropriate. The clarification of limitations (e.g., residual confounding, medication adherence, subgroup rationale) enhances the transparency and integrity of the findings. The authors provided clear explanations regarding the observed benefits of vasodilating beta-blockers in subgroups such as CKD and elderly patients. The clinical message is relevant, especially in the context of real-world use of beta-blockers post-AMI in patients with mildly reduced LVEF. I recommend acceptance of this version.

- Thank you for your thorough review and encouraging feedback on our revised manuscript. We are grateful for the constructive suggestions that have helped us improve the clarity and quality of our work.

Reviewer #2:

Thanks to the authors for the great effort addressing our comments.

Please try to include the data used in your responses as a part of the results. Example, you showed data on LVEF longitudinal changes as a part of your responses to the reviewers, but that data is not in the manuscript, and seems to be valuable.

- We sincerely thank you for the valuable comments and kind words. As suggested, we have now incorporated the LVEF longitudinal change data into the Results section of the revised manuscript:

Follow-up echocardiography was available in 41% of patients at 1-year, 19% at 2-year, and 18% at 3- year. Among those with follow-up data, the proportion of patients showing a ≥10% improvement in LVEF at 1 year was slightly higher in the vasodilating beta-blocker group compared to the conventional beta-blocker group (13.4% vs. 11.6%; P=0.104 in the entire cohort, 14.2% vs. 13.2%; P=0.331 in the PSM cohort), though these differences were not statistically significant. Similar trends were observed at 2-year (7.8% vs. 7.2%; P=0.554 in the entire cohort, 6.3% vs. 5.8%; P=0.430 in the PSM cohort) and at 3-year (5.9% vs 5.6%; P=0.716 in the entire cohort, 5.8% vs. 5.1%; P=0.538 in the PSM cohort). The proportions of patients who experienced a decline or minimal changes (0–10%) in EF were also comparable between groups. Overall, there were no statistically significant differences in LVEF recovery between vasodilating and conventional beta-blockers in either the entire or PSM cohorts.

Reviewer #3:

Thank you for answering my queries. No further questions. While the paper still has some limits, most of the important issues have been dealt with.

- We appreciate your thoughtful feedback throughout the review process. We are pleased to hear that the major concerns have been addressed to your satisfaction. Although certain limitations remain, we are grateful for your recognition that the key issues have been adequately dealt with. Thank you for your time and valuable input.

Reviewer #4:

The authors have adequately addressed the queries from reviewers and taken care to modify the text of the article to reflect areas of uncertainty raised by reviewers.

- We sincerely thank you for the constructive comments and positive evaluation. We are glad that the revisions and clarifications made to the manuscript have adequately addressed the concerns raised. Your recognition of our efforts to reflect areas of uncertainty is much appreciated.

---

## [Editor Report · Decision Letter 2]

Long-term benefit of vasodilating beta-blockers in acute myocardial infarction patients with mildly reduced left ventricular ejection fraction

PONE-D-25-05797R2

Dear Dr. Lee,

We’re pleased to inform you that your manuscript has been judged scientifically suitable for publication and will be formally accepted for publication once it meets all outstanding technical requirements.

Kind regards,

Ahmed Qasim Mohammed Alhatemi, Mbchb

Academic Editor

PLOS ONE

---

## [Editor Report · Acceptance letter]

PONE-D-25-05797R2

PLOS ONE

Dear Dr. Lee,

I'm pleased to inform you that your manuscript has been deemed suitable for publication in PLOS ONE. Congratulations! Your manuscript is now being handed over to our production team.

Kind regards,

on behalf of

Dr. Ahmed Qasim Mohammed Alhatemi

Academic Editor

PLOS ONE